# Reducing the Impact of Influence Factors on the Measurement Results from Single-Coil Eddy Current Sensors

**DOI:** 10.3390/s23010351

**Published:** 2022-12-29

**Authors:** Sergey Borovik, Marina Kuteynikova, Yuriy Sekisov

**Affiliations:** Samara Federal Research Scientific Center RAS, Institute for the Control of Complex Systems RAS, 443020 Samara, Russia

**Keywords:** single-coil eddy current sensor, measuring system, powerplant, state monitoring, influence factor, multidimensional displacements, complex shaped object, temperature effect compensation

## Abstract

Single-coil eddy current sensors (SCECS) form a separate and independent branch among the existing eddy current probes. Such sensors are often used for aviation and aerospace applications where the conditions accompanying the measuring process are harsh and even extreme. High temperatures (up to +600 °C in the compressor and over +1000 °C in the turbine of gas turbine engines), the complex shape surfaces of the monitored parts, the multidimensional movement of the power plants’ structural elements, restrictions on the probes number and their placement in the measuring zone are the main factors affecting the reliability and accuracy of the measurement results obtained by the sensors. The article provides an overview of the relevant approaches and methods for reducing the impact of influence factors on the measurement results from SCECS based on the extensive experience of more than 30 years of research and development being carried out in the Institute for the Control of Complex Systems of Russian Academy of Sciences. The scope of the solutions discussed in the article is not limited to SCECS measurement systems only but can also be extended to the systems with primary transducers of other designs or other physical principles.

## 1. Introduction

As it is known, the eddy current probes (ECP) are widely used as a part of measurement and control systems for industrial automation, transport systems, aviation and space technics, etc. [1,2,3,4,5,6,7,8]. The ECP for aerospace applications [9,10,11,12,13,14,15,16,17,18,19,20,21] is of particular interest among such sensors. These ECPs are usually developed to operate under harsh and even extreme conditions primarily associated with high temperatures in the measuring area reaching +1000 °C or more in the turbines of modern gas turbine engines (GTE). To expand the ECP operation temperature range, the forced liquid cooling of the probes [1,15] and making the ECP working coils of platinum by placing the coils on a ceramic frame [11,13,14] are used. However, these technologies are expensive and are not always applicable in practice (for example, water cooling is extremely problematic in research and testing of aircraft and rocket engines).

Another type of high-temperature ECP is the single-coil eddy current sensors (SCECS) with sensitive elements (SE) of the simplest geometrical shape in the form of a single current loop or its part (a segment of a linear conductor) [9,18,19,20,21] which constitute a separate and independent branch among the existing probes operating on the eddy current principle. The simple design of the sensor’s SE has high reliability, and technological effectiveness in manufacturing and does not require the use of high-temperature inter-turn insulation. The use of materials with a high melting point (for example, the same steel alloys that are used in turbine blades) for the manufacture of the SE and other sensor’s structural elements, ensures the required mechanical and thermal resistance of the sensor and its operability under extreme temperatures in GTE gas–air path without additional cooling of the SE.

At the same time, the self-inductance of a single-coil SE is extremely low (of about a few units, tens of nH) and the useful changes in the SE’s inductance from the influence of the measured parameters are only about 1…2% of its self-inductance. The direct connection of the SE in the measuring circuit (MC) of a secondary converter is carried out through the matching transformer (MT) which is moved out of the high-temperature zone through the current leads with minimal self-inductance. Nevertheless, considering the small changes in the SE’s inductance from the influence of measured parameters, ensuring an acceptable signal-to-noise ratio (SNR) when converting the output parameters of the SCECS into an electrical signal is not a trivial task, especially in EMI-heavy environments during the sensors’ operation on the real objects [22,23,24].

The features of the machine design and its operation specifics are also significant factors that influence the process of information obtaining about the monitored power plants’ parameters with the help of SCECS. For instance, in many practical applications, the measurement task is formulated in such a way that only one coordinate of the structural element displacement is monitored. The radial clearance measuring between the impeller blade tips and the stator of the compressor or turbine of the aircraft GTE [9,10,11,12,13,14,15,16,17,18] and the wear diagnostics of the combined journal-and-thrust bearing of the liquid-propellant rocket engine’s turbo-pump unit by measuring the shaft axial movement in the bearing [19] are the typical examples of such tasks. However, the changing temperatures, centrifugal, thrust, and aerodynamic loads in real operating conditions of the propulsion system cause the axisymmetric and asymmetric deformations of the engine’s structural elements [25] resulting in a complex multi-dimensional movement of the monitored object (blade tip, tooth of the measurement disc, etc.), on several coordinates. The informative parameter of the SCECS’s SE (the inductance) responds to all coordinate component changes. Therefore, the object displacements in all the coordinates except the monitored are inconvenient and lead to additional measuring errors.

Finally, it should also be noted that, regardless of the ECP design, the temperature effect on both the elements of the sensor itself and the object with which electromagnetic interaction occurs is another interfering factor that has a significant impact on the transducers’ metrological characteristics. Additionally, such influence is enhanced as the ECP operation temperature range increases. This is particularly important in the tasks of monitoring the state of aircraft and rocket engines’ elements. As it is noted in [26] with reference to the study [27], the temperature changes in the SCECS’s equivalent inductance (sensor’s informative parameter) in the temperature range from +20 °C to +1000 °C are five times higher or more than the changes in the same parameter determined by interaction with the monitored object.

With over more than 30 years of history of the creation of SCECS and measurement systems based on them, the Institute for the Control of Complex Systems of Russian Academy of Sciences has accumulated extensive experience if not complete elimination, then a significantly reducing impact of these (and not only) influence factors on the measuring results. The aim of the article is to provide an overview of the most relevant approaches and methods which are both circuit and algorithmic in nature. The aim determined the logic and the structure of the article which has seven sections, in addition to the introduction and the conclusion. The main part of the article begins with a description of SCECS and its operation which is necessary for further understanding of the proposed solutions. Then, one of the possible formal classifications of the factors influencing the results of the transformation of the parameters measured by SCECS is given in Section 3. Further sections consistently provide an explanation of the ways for reducing the impact of the identified influencing factors.

## 2. SCECS: Typical Design and Principle of Operation

There currently exists many varieties of SCECS [9,18,19,20,21,22,23,24,25,26], but they all can be divided into two main groups:SCECS with SE as the whole current-carrying coil (circuit) (Figure 1a,b);SCECS with SE as a fragment of the coil (for example, a segment of a linear conductor) (Figure 1c,d).

Regardless of the design, the SCECS consists of three basic functional elements: SE, MT, and current leads. The ability of the sensor to operate under extreme conditions is ensured primarily by the SE’s simple design, which does not require high temperature insulation, and its mechanical and thermal strength is determined by the properties of the material used only. For example, if the SE is made of heat-resistant steel alloy, like the one used in aircraft engine building (e.g., BZh98, EI868, XH60BT [28]; Inconel 600, Inconel 625 [29]), it retains its structural strength at temperatures above 1000 °C and ensures the sensor’s efficiency under extreme conditions in the GTE gas–air path.

The SE is placed into the measurement zone by means of current leads, which are usually made by “non-inductive” technology in the form of coaxial cylinders or closely spaced metal strips. The current leads provide the MT removal from the zone with high temperatures and connect the SE of the sensor with the secondary winding of the MT in the form of a “volume coil” that is electrically connected to the MT primary winding. To minimize the sensor’s size its MT windings are placed on the core, which is made of high-frequency ferrite with high magnetic permeability. If the temperature conditions in the measurement zone are acceptable for the magnetic circuit of the MT, then it is no longer necessary to use current leads and in the extreme case, the SE can be directly connected to the MT’s secondary winding [9,30,31].

The principles of SCECS functioning are reflected with different degrees of detail in publications [18,30,32,33,34,35]. However, an understanding of the proposed solutions in the article is difficult without a brief explanation of the sensor’s operation. The simplest double contour model of the electromagnetic interaction between the SE of the SCECS and the electrically conductive object is presented in Figure 2a. The time diagrams of changes in the currents and equivalent inductance of the SE on the example of the measuring of radial clearances between the blade tips and the stator of a power plant are presented in Figure 2b.

The monitored object (e.g., blade tip) is located at a distance *y* from the SE (Figure 2a). The sensor is powered by the rectangular voltage pulses with amplitude *E*. Let us assume that the MT does not distort the front edge of the supply voltage, which excites the growing current *i_SE_* in the SE circuit (*i*_1_ is a current in the primary MT winding, *i_obj_* is an eddy current in the contour that imitates the target).

When the monitored object is at a large distance from the SE (*y* → ∞), there is no electromagnetic interaction between the SE and the target (mutual inductance *M* = 0 and *i_obj_* = 0). If the influence of eddy currents in the SE and other sensor’s elements associated with the magnetic field caused by *i_SE_* can be neglected, the current *i_SE_* is determined only by the self-inductance *L_SE_* at the beginning of the transition process. In this case, the *i_SE_* changes in time will have an exponential growing character [18] (Figure 2b, dashed curve) and the SE’s equivalent inductance will equal its self-inductance (*L_s_* = *L_SE_*).

As the monitored object approaches to the SE (*y* → 0), the eddy currents appear in the target under the action of a magnetic field created by *i_SE_* and time-varying *i_obj_* current appears in the contour that imitates the target. The current *i_obj_*(*t*) affects the resulting magnetic field and this leads to the changes in the shape of the current *i_SE_*(*t*) and its deviation from the exponential dependence (Figure 2b, solid curve). Such deviation can be interpreted as the influence of a time-variable equivalent inductance *L_s_*(*t*), whose instability in the transition mode is explained by the influence of the eddy current *i_obj_*.

## 3. Factors Affecting SCECS: Types and Features of the Influence

Figure 3 presents a possible formal classification of the factors influencing the results of the transformation of the parameters measured by SCECS. Of course, this classification does not exhaust all possible factors, but it gives an overview of the main ones that can be called decisive.

As can be seen in Figure 3, all the variety of factors affecting SCECS can be divided into two large groups:Factors due to the specifics of the monitored object and the peculiarity of the measurement problem being solved.Factors caused by external or (and) internal electromagnetic radiation on the sensor’s parts.

A more detailed analysis of these groups of factors and an assessment of the degree of their influence on the measurement result from SCECS are given below.

### 3.1. Influence of the Specifics of the Monitored Object

The first group of influence factors is the most numerous and determines the additional errors of SCECS-based measuring instruments associated both with a change in the electrophysical characteristics of the sensor’s parts due to changing environmental conditions in the measurement zone, and with a deviation of the measurement conditions from those that were at metrological certification of systems’ measuring channels. In addition, here first and foremost, it goes about temperatures, which can vary in practical applications over a wide range from −60 to +1000 °C and higher [13,15,18,30]. The changes in geometric dimensions, electrical conductivity, and magnetic permeability of the SCESC’s SE and the monitored object materials lead to a significant change in the level of the sensor’s informative parameter (equivalent inductance), unrelated to the measuring task under these conditions. It should also be noted that SCECS’s output parameter is affected not only by high temperatures in the area where its SE is placed, but also by increased temperatures in the MT location zone. Most high-temperature SCECS use thermostable ferrites with a high Curie point value in MT (e.g., Mn-Zn ferrite core 700 NM with *T_C_* = +400 °C and initial permeability µ*_i_* = 700). However, if the temperature in the MT’s location zone exceeds the Curie temperature, the SCECS characteristics dramatically deteriorate.

The circuit methods coupled with active thermal compensation techniques are usually used to reduce the SCECS temperature sensitivity. Section 5 of the article describes these methods.

The dynamic loads affecting the power plant components in operation mode have also a significant influence on the conversion of the measured by SCECS technical state parameters of the monitored machines and mechanisms in addition to the high temperatures in the measuring zone. As it is noted in publications [18,19,27,30,32,35] devoted to the problems of radial clearances measuring in compressor or turbine of GTE and early wear diagnostics of turbo-pump units’ (TPU) thrust bearings by monitoring the TPU shaft’s axial displacement, the centrifugal, thrust, and aerodynamic loads lead to the complex multi-dimensional movement of the monitored constructive element (blade tip, tooth of the measurement disc, etc.), in real operating conditions of the propulsion system. In this case, the displacement of the material point selected at the monitored element has a fundamental vector character and is determined by several coordinates in the Cartesian reference system *OXYZ*, the center of which (point *O*) is rigidly attached to the SCECS installation location (geometric center (g.c.) of the sensor’s SE).

The multi-dimensional movement of the monitored objects has a significant impact on the output signal of the SCECS, which integrally contains information about all coordinates of such displacement. So, with reference to the examples above, the components of the blade tips displacement along the *X* and *Z* axes (axial displacement of the shaft and bending of the blade airfoil in the axial direction and in the direction of the wheel rotation) will be the interfering factors which make it difficult to obtain information about the measured radial clearance in GTE compressor or turbine (movement in the direction of the axis *Y*) [18]. Similarly, the radial displacement of the tooth of the measuring disk due to elastic and thermal deformations of the disk is the factor that makes it difficult to measure the shaft’s axial displacement in the TPU thrust bearing (displacement along the *X* axis) [19].

Measuring the displacements of the monitored object along all coordinates that affect the SCECS’s output signal is possible by using so-called “cluster measurement methods” [32,33] realized by means of groups of identical sensors (clusters) whose SE are oriented in a certain way toward the object and the number of sensors in the cluster corresponds to the number of monitored coordinates. Such methods are discussed below in Section 6 of the article.

It should be separately noted that the need to ensure a high-precision synchronization of the inductances conversion of all SCECS in the cluster with the specified location of the monitored object relative to the sensor is another influential factor related to the dynamics of the object according to the existing cluster methods of determining the multi-coordinated movements of power plants structural units with a discrete surface (blade tips of compressor or turbine impellers, gear teeth, etc.) [18,27,30,32,35,37,38,39,40]. In [41], it is noted that in many practical applications the industrial RPM sensors (e.g., DCHV-2500 [30,32], IS-445 [19]) are traditionally used for these purposes. Unfortunately, it is not always possible to arrange the required mechanical connection of the RPM sensor with the shaft, and especially to duplicate it if the parallel measurement of rotation speed and synchronization are necessary. In addition, the PRM sensors usually provide a measurement of the “average” rotation speed, which is acceptable in the steady operation mode of the power plant, but is unacceptable in variable load modes associated, for example, with sharp revolutions set or drop.

The design features of the monitored object also affect the results of the transformation of the parameters measured by SCECS. In particular, the location of adjacent power plants’ structural elements (blades of the compressor or turbine impellers, gear teeth, etc.), can be linearly commensurate with the length of the sensor’s SE. As a result, the neighboring electrically conductive parts will affect the useful changes in the SE equivalent inductance and this influence can be very significant. In addition, the complex surface shape with which the electromagnetic interaction of the SCECS’s SE occurs is also an interfering factor, and its impact not only makes it difficult to obtain reliable data but may also cast doubt on the operation capacity of previously developed methods. For example, blades with a high curvature of the surface in their end part (the cross-section by the plane perpendicular to the blade’s axis has a sharply expressed «crescent» shape, and the cross-section of the blade’s end part by a plane parallel to the blade’s axis is «*U*-shaped») are used in existing and prospective turbine designs. This curvature and the complex shape of the blade means that the possible mutual locations of the SCECS’s SE and the monitored blade tip cannot be entirely determined in advance before the test starts. Therefore, the measurement information can be obtained under conditions that differ from those in which the measuring system was certified.

Finally, the feature of the SCECS placement on a monitored object is a significant influence factor, too. The limited space for sensors installed on the power plant body, for example, makes it impossible to place the required number of SCECS in the cluster. Therefore, it is necessary to abandon the measuring of separate coordinate components, and this is, as already noted, the cause of additional errors.

### 3.2. Electromagnetic Influence on SCECS Constructive Elements

The SCECS operation on real power plants is almost always accompanied by a rather complex electromagnetic environment related to the presence of a large number of simultaneously operating radio-electronic, electrical measuring, electromechanical and other devices that generate electromagnetic oscillations in the surrounding space, as well as the presence of natural sources of electromagnetic radiation [42].

It should also be noted that the eddy currents excited by the SE current are generated not only in the monitored structural element, but also in all parts of the SCECS design, including the SE itself. Eddy currents in the sensor parts and in the monitored object affect not only the SE circuit of the SCECS, but also interact with each other in accordance with the principle of “each with each”. As a result, the volume distribution of eddy currents over the sensor changes in time, and as a result, the sensor informative parameter (equivalent inductance) also changes in time when the object is in the same position relative to the SE of the SCECS.

It is obvious that «external» and «internal» electromagnetic fields have an impact on the operation of any ECP. However, as it is noted in the introduction, the self-inductance of the SCECS SE is of about a few units, tens of nH, and its changes during the electromagnetic interaction of the sensor and the monitored object are negligible (about 1–2%). Under these conditions, the electromagnetic effect on SCECS parts is critical regardless of the nature of the radiation, and special measures to ensure an acceptable SNR are required.

## 4. SCECS Signals Conversion with Acceptable SNR and Uninformative Parameters Suppression

The pulse power supply (see Section 2) is used as an effective way to increase the level of the SCECS useful signal. Contrary to the traditionally applied in the ECP high-frequency harmonic power supply [1,3,43], the pulse mode allows us to increase the amplitude of the information signal at the MC output by increasing the voltage of the SCECS power pulse while maintaining the average power dissipated in the sensor. The thermal mode of the sensor is achieved by reducing the pulse duration [44].

The signal at the MC output also has a pulsed nature at a pulsed SCECS power supply. It greatly simplifies further switching and analog-to-digital conversion. In addition, the non-informative parameters reduction effect can be obtained in MC by choosing in a certain way the ratio between the pulse duration and the MC time constant [45] and, in particular, the active resistance suppression, which varies widely from the temperature influence in the measuring zone.

It is noted in [28] that the beginning of the transition process at the time of formation of the read pulse (*t* → 0) is the most attractive time point for SCECS signals transformation. As it is known, the method of the first derivative [45] (one of test transition methods) provides the minimum duration of the pulse of a power supply among the pulse techniques used for converting the parameters of the inductive and eddy current sensors. Figure 4 presents an equivalent scheme (Figure 4a) and time diagrams (Figure 4b) explaining the first derivative method (without taking into account the self-capacitance *C_s_* of the SCECS).

The SCECS in Figure 4a is represented by its equivalent parameters—inductance *L_s_*, resistance *R_s_*, and capacity *C_s_*. The equivalent inductance of the sensor (*L_s_*) is associated with the equivalent inductance of the SE (*L_se_*) by the expression $L_{s}={n_{t}}^{2}\cdot L_{se}$, where *n_t_* is the MT transformer ratio determined by the inductances of its primary and secondary windings ($n_{T}=w_{1}/w_{2}$, *w*_1_ and *w*_2_—the number of turns, and *w*_2_ = 1) [28]. In addition to the SCECS, the circuit (Figure 4a) contains a switching power supply (SPS) with electromotive force (emf) *E* and internal resistance *r*, and a current differentiation device (DD) with a zero input resistance.

If the self-capacity of the MT primary winding (*C_s_*) is not taken into account, then the transition process in the circuit with pulsed supply *E* and time-varying of *L_s_*(*t*) is described by the equation:(1)$$L_{s}(t)\frac{di}{dt}+i\cdot \left[\frac{dL_{s}(t)}{dt}+R_{s}+r\right]=E.$$

Since *L_s_*(*t*) is a continuous function, the value of the current derivative can be found without solving the Equation (1) for zero initial conditions at the moment of the voltage *E* generation (at *t* → 0 current *i* = 0):(2)$$i\cdot \left[\frac{dL_{s}(t)}{dt}+R_{s}+r\right]=0,\;{\frac{di}{dt}|}_{t\to 0}=\frac{E}{L_{s}(0)},$$
where *L_s_*(0) is the inductance at *t* → 0. This means that the current derivative does not depend on the sensor’s resistance *R_s_* and is determined by the instantaneous value of the inductance.

Unlike the idealized circuit, the self-capacitance (*C_s_*) of the SCECS shunts *L_s_* and *R_s_*. Therefore, the current *i*(*t*) is determined by the capacitive component at the beginning of the transient process and by the inductive component when the capacitance is charged. In this case, the derivative reaches its maximum not at *t* → 0, but after some time *t_m_* that does not exceed several tens of nanoseconds. The area of the transient process, where the current *i* is determined by the capacitive component has an irregular character and is not suitable for the converting of the inductance. Therefore, the maximum of the derivative at the beginning of the regular area of the transient process (*t* = *t_m_*), where the influence of the main parameter of the circuit becomes decisive, is taken as an output signal. The conversion process can be repeated after the derivative is fixed and the energy in the circuit is dissipated.

## 5. Reducing the Temperature Effect on the SCECS Informative Parameter

Traditionally the SCECS-based systems for measuring and monitoring the state of complex technical objects use the circuit methods to reduce the additional temperature error of the sensors. The methods imply the elimination of the temperature impact by using additional temperature-dependent signals. In particular, the differential MC with two identical SCECS parameters in the adjacent arms is used to convert the changes in the SCECS SE equivalent inductances during their interaction with the monitored object. The first SCECS performs working functions and directly interacts with the monitored object. The second SCECS (SCECS witness) performs compensatory functions and is installed in a separate mounting hole on the object’s body. The SE of witness SCECS (SE of SCECS_2_, SE_2_) is placed in the measurement zone in such a way that the temperature conditions for it and for the SE of the working SCECS (SE of SCECS_1_, SE_1_) would be identical, but the electromagnetic interaction between SE_2_ and the monitored object would not occur.

Figure 5 presents two versions of the most frequently used differential MC which implement the first derivative method and are implied for SCECS parameters conversion.

The first MC (Figure 5a) contains the Blumlein transformer measuring bridge [46] whose two adjacent arms include SCECS_1_ and SCECS_2_ with sensing elements SE_1_, SE_2_, and matching transformers MT_1_, and MT_2_. The sensors are powered by short rectangular voltage pulses with a frequency of up to several megahertz from an SPS. The feed pulses are formed by the contactless key elements (*K*_1_, *K*_2_) in the DC voltage source (*E*). The key elements and the DC voltage source are shunted by the *R_d_*_1_ and *R_d_*_2_ resistors through which the energy stored in the differential circuit is dissipated during the time of the feed pulse (the key elements *K*_1_, and *K*_2_ are open when the energy is dissipated). Magnetically coupled inductors (the coupling coefficient is equal to one) are used as a current DD. In this case, the DD’s input resistance in relation to clamps *a*, and *b* is determined only by the active resistance of the inductance coils.

It is shown in [30] that when the monitored object moves relative to the SE only of a single SCECS, as it just happens in the case under consideration, the voltage at the MC output at the time moment *t* → 0 is determined by the expression:(3)$$U_{out}|{}_{t\to 0}\approx E\frac{\Delta L}{L_{0}},$$
where *L*_0_ is the self-inductance of the SCECS (to simplify it is assumed that SCECS_1_ and SCECS_2_ are completely similar in their parameters and therefore *L_s_*_1_ = *L_s_*_2_ = *L*_0_); Δ*L* is the change in the SCECS inductances caused by the movement of the target in the sensitivity zone of the working sensor. So, the Blumlein bridge provides an almost linear conversion, and the instability of the inductance *L* does not have a significant impact on the result [47].

The MC variant in Figure 5b uses the operational amplifier (op-amp) as a differentiating device instead of inductance coils with close magnetic coupling [30,32,33]. The circuit contains a non-equilibrium bridge with MT of both SCECS (MT_1_ of SCECS_1_ and MT_2_ of SCECS_2_) in its adjacent arms and the current-to-voltage converters (CVC) on the base of op-amps with resistors *R* in the feedback circuits. The SPS is included in the bridge’s power supply diagonal and forms a brief pulsed voltage like in the previous MC variant. The CVC convert the currents *i*_1_(*t*) and *i*_2_(*t*) into voltages *U*_1_(*t*) and *U*_2_(*t*) at their outputs [48]. At the same time, the CVC allows the maintenance of the currents in SCECS and the time constants of the MC like in the Blumlein bridge with a similar power supply. That is why in [49] this MC was named the «electronic analog of the Blumlein bridge».

The different voltage from the CVC outputs (Δ*U* = *U*_1_ − *U*_2_) is either differentiated by the op-amp [50] or the so-called operation of “approximate differentiation” is carried out. In the last case, the difference voltage is fixed after a short time interval equal to the duration of the feed pulse and then the analog-to-digital conversion with preamplification or without it is produced [51,52]. In the last case, DD is not used and the output voltages (*U*_1_, *U*_2_) are delivered directly to the differential inputs of the analog-to-digital converter (ADC). This ensures that the digital code at the ADC output will be proportional to the voltage difference Δ*U* at the end of the pulse of the power supply.

It should be noted that the use of the microelectronic base creates a perspective for the MC’s miniaturization and its integration in the SCECS or embedding into the communication line at a short distance from the sensor. Moreover, the integration of MC with ADC and microcontroller-based computing devices within a single design allows us to create so-called “intelligent sensors” [53,54] with automatically performed self-control and advanced information processing.

Unfortunately, the differential MC makes it possible to reduce the temperature effect on the sensor, but not to remove it. The main reason why the complete elimination of the temperature effect is not feasible is the technological impossibility to manufacture two absolutely identical SCECS. In addition, in most practical applications when the monitored object moves relative to the SE of only one sensor (e.g., measuring of the radial clearances in GTE), the result of the electromagnetic interaction between the SE of the working SCECS and the monitored object (blade tip) depends not only on the temperature of the sensor’s elements, but also on the temperature (more precisely, temperature changes in electrophysical parameters) of the target [55].

To further reduce the temperature effect on the SCECS the active temperature compensation methods are used. The methods involve the direct temperature measuring in the SE location zone with further algorithmic correcting of the temperature effect on the SCECS and the monitored object [30]. For this purpose, the SCECS is completed with at least one thermocouple and a hot junction is placed next to the sensor’s SE and provides the temperature measurement in the working area [18,30]. In fact, the algorithmic temperature correction involves the experimental obtaining of the calibration characteristics of the measuring channels with SCECS in the form of dependences of voltages at the MC output (more often codes after analog-to-digital conversion of the MC output signals) on the measured parameters, considering the temperature changes θ in the SE locating area. Such studies are carried out using special calibration devices with a thermal chamber and a fragment of the target just before the sensors are installed on the monitored object or test bench [30]. A variant of such calibration devices, designed for metrological certification of the channels with SCECS of the systems for radial clearances measuring in GTE, is shown in Figure 6.

The device contains a cylindrical vertically located electric thermal chamber 1 inside which a fragment 2 of the monitored object (blade, fragment of a blade wheel, etc.), or its imitator is placed. The SEs of SCECS_1_ and SCECS_2_ are entered into the heat chamber and fixed in bushings 3, the same as those installed on the body of the monitored object. This ensures the identity of the conditions during the sensors’ calibration and their operation on a real object. The bushings are fixed on the platform 4 located above the top cover of the heat chamber and pass inside it. A fragment of the monitored object (in this case, the blade) is placed on the upper end of the ceramic rod 5, which passes through a slot at the bottom of the heat chamber. The lower end of the rod is fixed on a mechanism that ensures the movement of the object fragment in the vertical and horizontal directions. The vertical movement is set by the handwheel of the micrometric screw 6 and is evaluated by the dial indicator 7. The micrometric screw is fixed on the carriage 8, and mounted on horizontal guides. Movement in the horizontal direction is set by the handwheel 9 and is controlled by a dial indicator 10. The heater of the thermal chamber is connected to the AC network 220 V, 50 Hz through a thyristor power controller. The regulator switch 11 allows setting ten fixed temperature values in the thermal chamber in the range from normal laboratory temperatures to +1100 °C. Temperature monitoring in the heat chamber is carried out using a thermocouple and a millivoltmeter. Each new calibration experiment begins with the establishment of the “zero position” when the sensor’s SE touches the surface of the target. In this case, the result distortion due to the influence of the thermal expansion of the fastening elements of the sensor and the target is eliminated. The touch recognition is confirmed by the LED indicator 12.

The sequence of operations related to obtaining the families of the calibration characteristics of the measuring channels with SCECS considering the analog-to-digital MC output signals conversion can be represented by the following verbal algorithm:The SCECS (or the cluster (group) of SCECS) together with a fragment of the monitored object (the target) are installed in a heat chamber in accordance with using measurement method.The specified temperature regime (temperature control point) is set and maintained in the heat chamber with the help of a temperature controller.The starting point (“zero position”) of the calibration characteristic is set by the moment when the SE of the working SCECS touches the target.The SCECS and the target move relative to each other by the handwheels 6, 9*,* and a multiple fixation of the digital code corresponding to the voltage on the output of the MC with SCECS in given mutual positions of the sensor’s SE and the target is made.Steps 2–4 are repeated for all temperature control points.

In general, the families of calibration characteristics formed in this way represent the multidimensional arrays of digital codes for the specified (usually in Cartesian coordinates *x*, *y*, *z*) mutual positions of the SCECS’s SE and the monitored object at a fixed temperature:(4)C=(Ck1i,…,kχj,θl)∈ℤk1m×…kχn×θp,
where *k*_1_, …, *k*_χ_ are possible coordinates of the monitored object displacement relative to the SE of working SCECS; θ—the temperature; *m*, …, *n*—the numbers of control points for each coordinate; *p*—the number of temperature control points. The families of calibration characteristics (4) are either directly stored in the measurement system’s memory in the form of interpolation tables or approximated by polynomial functions of several variables and only the corresponding coefficients of the polynomials are stored [32]. In the last case, the families of calibration characteristics for *i*-th SCECS can be written as
(5)$$C_{i}=f_{i}(C_{k1},C_{k2},\ldots ,C_{k\chi },\theta ).$$

In the normal operation mode of the measurement system the desired position of the monitored object is calculated on the basis of the current code values corresponding to the changes in SCECS inductances and the temperature in the SE location zone using calibration characteristics (4). It should be noted that when the target moves at several coordinates (χ > 1) the use of cluster measuring methods is needed to determine the monitored object’s position in the given coordinate system [18,32,33]. These methods, as well as the methods for coordinates calculating, will be discussed below.

It is obvious that the experimental obtaining of the calibration characteristics is associated with a high complexity of the process, even with its automation [56]. The high labor intensity of the metrological experiments can be reduced by minimizing the number of points (increasing the step between the adjacent samples) at which the calibration characteristic is measured. Considering the fundamentally non-linear nature of the dependence of the equivalent inductance of the SCECS (and, accordingly, the MC output signal) on the measured parameter, the possibilities of this method of reducing the labor intensity of the calibration are limited by potentially large errors in defining the measured parameters. The alternative approach is proposed in [57,58]. It assumes the rejection from the experimental acquisition of the families of calibration characteristics and their replacement by families of the conversion functions in the form of the similar dependences of voltages (codes) at the output of the MC with SCECS on the measured parameters at different temperatures. The computer models of electromagnetic interaction between the SCECS’s SE and the target considering the further conversion of the equivalent parameters of the sensor in the MC are used to calculate the conversion functions [59,60,61,62]. The value of the final temperature error of the measuring channel with SCECS in this case is determined by the adequacy of the used models of the sensor and the MC.

It should also be noted that the design and technological limitations do not allow the thermocouple to be placed directly on the SCECS’s SE. As a result of the removal of the thermocouple from the SE along the current lead axis (even for a small distance), the temperature at the thermocouple installation point will differ from the temperature of the sensor’s SE, which is often placed in the flow of the heated moving medium of the monitored object (gas–air medium in the GTE compressor or turbine, oil flow in the lubrication system of the power plant, etc.) At the steady heat fluxes, the relationship between the SE temperature and the temperature at the point of the thermocouple installation is considered during calibration. In the case of a rapidly changing temperature (e.g., variable engine operation modes) a thermal transient process will occur in the SCECS current leads and the temperature changes at the thermocouple installation point will be delayed relative to the SE temperature changes. This will cause the voltage of the MC with SCECS to change faster than the correction algorithm will “restore” it. For a more accurate determination of the SE temperature both in stationary and transient thermal modes the thermal sensor model is proposed in [63]. The model uses the results of direct temperature measurements at two points in the SCECS body. For this purpose, an additional thermocouple is installed in the sensor’s MT location area. The same literature source [63] shows that the use of a thermal model makes it possible to reduce the SCECS additional temperature error in the transient mode to 1.1%.

In addition to the SE «temperature recovery», the additional thermocouple in the SCECS body performs the MT ferrite core temperature monitoring to prevent the exceeding of the Curie point during the sensor’s operation. If it is not possible for some reason to provide an acceptable MT temperature mode, then the forced air cooling of the MT should be used. As an alternative, the construction of the SCECS with so-called «air» MT without ferrite core is proposed in [33]. The frame of the MT primary winding of such SCECS is made of high-temperature dielectric material capable of operating in extreme temperatures. At the same time, the use of «air» MT is accompanied by a loss of the sensor’s sensitivity. The compensation of the specified effect requires greater MT and sensor dimensions.

## 6. The Impact of Target Multidimensional Movement

It was noted in the analysis of the factors influencing SCECS (Section 3), that the monitored constructive elements of the machines and mechanisms in real operating conditions perform complex multidimensional movements associated with factors of different physical nature. At the same time, only one of the coordinates of such movement (e.g., radial clearance, wheel displacement along the shaft axis, etc.), in the selected reference system (often the Cartesian *OXYZ* coordinate system, the center of which (point *O*) is rigidly attached to the SCECS location on the object) is usually monitored. In this case, the components of target movements along the other coordinates refer to interfering factors influencing the output signal of the measuring channel with SCECS, which must be considered when determining and calculating the monitored parameters.

Measuring the multi-coordinate displacements of the monitored objects is provided by using “cluster methods” based on groups of identical SCECS whose SE are oriented in a certain way toward the target and the number of sensors in the cluster corresponds to the number of monitored coordinates χ [18,32,33]. The desired coordinates (*k*_1_, *k*_2_, …, *k*_χ_) of the object’s movements, both monitored and unmonitored, are calculated on the results of measurements in a closed series provided by all SCECS in the cluster [32]. For this purpose, the system of Equations (6) is solved. Each equation in (6) is an experimentally obtained family of calibration characteristics (5) with specific values of digital codes (*C*_1_, *C*_2_, …, C_χ_) fixed at a given time corresponding to the calculated position of the monitored object in the sensors measuring area.
(6)$$\left\{ \begin{array}{l} C_{1}=f_{1}(k_{1},k_{2},\ldots ,k_{\chi },\theta ),\\ C_{2}=f_{2}(k_{1},k_{2},\ldots ,k_{\chi },\theta ),\\ \cdots \\ C_{\chi }=f_{\chi }(k_{1},k_{2},\ldots ,k_{\chi },\theta ). \end{array}\right.$$

The solution of the equation system (6) is usually carried out by Newton’s method [64,65], which provides iterative procedures that are characterized by quadratic convergence and relatively low computational costs. In this case, the families of calibration characteristics must be monotonic functions in the required coordinates’ ranges and must provide sufficient sensitivity. In addition, the independence of the equation system (6) and its solvability related to the desired coordinates must be ensured. However, it should be noted that real SCECS have local extremums on their calibration characteristics and therefore the use of Newton’s method brings a significant limitation of the operating range of coordinate measurements. In [33] a modified algorithm for two-coordinate displacement calculating is proposed. The algorithm uses non-monotonic segments of the calibration characteristics and allows us to expand the range of measured coordinates.

Despite all the variety, the cluster methods can be divided into two large groups:Methods that provide the SCECS concentration in close proximity to the observation point (the “concentrated cluster” of SCECS);Methods that provide the SCECS distribution on the monitored object (the “distributed cluster” of SCECS).

The methods of the first group were the simplest for implementation, and therefore historically the first. They are focused on the use of concentrated clusters of SCECS with the simultaneous (parallel) conversion of the SE informative parameters (inductances). The most popular variants of the methods in relation to the problem of the monitoring of the discrete surface (e.g., blade tips of a compressor or turbine impeller), including the so-called “degenerate cluster” with one SCECS [66], are presented in a schematic form by their SE in Table 1. Point *O* on the diagrams is the origin and geometric center (g.c.) of SCECS’s SE. The dimensions of the monitored object, SCECS, and the ratio between them are not considered in the diagrams. However, it is assumed that the conductor segments forming the SE do not cross. The clusters of SCECS that implement the compensation functions are also not shown in the diagrams.

The use of “concentrated cluster”-based methods is associated with certain difficulties and limitations, which in one way or another depend on the close sensors’ placement in a small area and on the design features of the SCECS itself. One of the main limitations is the need to make many mounting holes in the monitored object on a relatively small area. The number of mounting holes increases in the case of increasing the number of monitored coordinates and thus the corresponding number of sensors in the cluster composition increases, too. Additionally, given the need to use additional witness SCECS to compensate for the temperature effect on the sensors, the number of mounting holes is doubled. Even when using the so-called “cluster SCECS”, which combines several SCECS in a single body [31,67], the problem is not eliminated. The “cluster SCECS” is significantly larger in size than traditional SCECS and although the number of mounting holes for the cluster sensors decreases, their overall dimensions increase and this is often considered a critical limitation, too. In addition, the pulsed conversion of the SE inductances in the MC is carried out simultaneously for all sensors in the cluster (all SE in cluster SCECS). In this case, the SE mutual electromagnetic influence decreases the sensitivity to the monitored coordinates and, as a result, reduces the coordinates measuring ranges.

The methods of the second group based on a “distributed cluster” of SCECS allow us to eliminate the concentration of SCECS and corresponding mounting holes on a small area, as well as the SE mutual electromagnetic influence during the simultaneous conversion of their inductances. In this case, the first SCECS in the “distributed cluster” is placed in the same position as in the “concentrated cluster”, while the rest sensors are equidistantly shifted around the body of the monitored object. The variants of the methods based on the “distributed cluster” of SCECS intended for previously given tasks of blade tips multi-coordinate movements monitoring (Table 1) are presented in Table 2.

In particular, in the monitoring of 2D displacements of blade tips (over *x*,*y*-coordinates), SE_1_ is placed in the same position as in a “concentrated cluster” (Table 1, Row 2) and SE_2_ is equidistantly shifted by an angle 1.5Δψ_b_ (Table 2, Row 1), where Δψ_b_ is the angular pitch of the blades on the impeller (this corresponds to the shift of the g.c. and the reference system to the distance *OO*’). The *x*,*y* coordinates are measured in two phases. In phase 1 the root of blade 1 passes g.c. (point *O*) and SE_1_ performs the working functions (SE_1_-W) and SE_2_ performs the compensation functions (SE_2_-C). In phase 2, the root of blade 1 passes point *O’,* and SE_1_ reverses its functions from working to compensation (SE_1_-C) and SE_2_ and vice versa from compensation to working functions (SE_2_-W).

If 3D displacements of blade tips are monitored (Table 2, Row 2), then the “distributed cluster” must have three SCECS. In this case, the second SCECS (its SE, SE_2_) is still shifted by an angle 1.5Δψ*_b_* and the SE of the third SCECS (SE_3_) is shifted by an angle Δψ*_b_* from the SE_2_ (the g.c. (point *O*″) and the reference system are shifted by the same angle Δψ*_b_*). The conversion and fixation of digital codes are carried out at the moments when the root of the monitored blade (1) simultaneously passes points *O*, *O*′, и *O*″ during phases 1, 2, 3, respectively (Table 2, Columns 2, 3, 4). It should be noted that, in contrast to 2D displacement measurements (Table 2, Row 1), pairs of working and compensation sensors (in phase 1—pair SE_1_-W/SE_2_-C, in phase 2—pair SE_2_-W/SE_2_-C, in phase 3—pair SE_3_-W/SE_2_-C) are formed sequentially at each period of impeller rotation using the key elements of the MC switch.

Additionally, when the number of allowed mounting holes in the object’s body is less than the number of monitored coordinates and the corresponding number of SCECS in the “concentrated” or “distributed” cluster, the so-called “incomplete cluster” of the SCECS is used. In the “incomplete cluster” “unmeasured” coordinates are calculated using specially developed real-time models based on the current parameters of the engine regime and its environment [38,68]. For example, the use of an “incomplete cluster” of SCECS in combination with online modeling of the blade bend in the system for the radial clearances measuring between the propeller blade tips and the stator shell of the ducted propfan made it possible to reduce the number of SCECS to one sensor at each control point on a stator shell [69].

## 7. Specifics of the Monitored Object Design: Impact and Ways to Reduce It

The desire to increase the efficiency of modern aviation and space equipment leads to the fact that parts of complex geometric shapes are often used in product components, and the products themselves have a complex configuration. An example is the blades of high surface curvature and complex torsional shape used in the existing and prospective GTE turbines. In addition, the adjacent elements of the product structures are located often near the monitored object and are also involved in the process of electromagnetic interaction with sensors’ SE. In relation to the example under consideration, due to the small step of blades installation on the impeller of a compressor or high-pressure turbine, the SE of SCECS is affected not only by the blade in its close proximity, but also by adjacent blades. These circumstances are additional influencing factors that not only make it difficult, but in some cases preclude the SCECS application. Possible ways to reduce the influence of the factors associated with the design features of the monitored object are given below.

### 7.1. Consideration of the Complex Surface Shape of Monitored Objects

The main problem of complex-shaped object monitoring is the multidimensional nature of their movements in the SCECS sensitivity zone. In this case, the electromagnetic interaction of the sensor’s SE will occur with different parts of the object’s surface varying in their geometric parameters (height and width) at each new time moment. Thus, the SCECS information parameter (inductance) will vary within a wide range even if the mutual positions of the sensor and the monitored object remain unchanged.

Based on the analysis of [17,33,35,41] it can be stated that the main way to take into account the complex surface shape of the monitored object is to select the SCECS’s installation location and the orientation of its SE in such a way that the specified features of the monitored object either do not affect the result of sensor’s output parameters conversion or can be considered in the subsequent processing of sensor’s signals. The application of computer models of the electromagnetic interaction between SCECS’s SE and the object [59,60,62] is highly effective in this case. The models allow at least to narrow the search space for acceptable solutions.

The end part of the turbine blade is given in Figure 7 as an example of a complex-shaped object. The view of the blade tip from the stator side is shown in Figure 7a and its cross-sections (A-A and B-B) are presented in Figure 7b. The protrusions in the blade’s end part are *U*-shaped. The distance between the protrusions depends on the blade thickness and decreases from the inlet edge to the outlet edge, as well as the curvature of the blade airfoil. The conditional blade’s geometric center is located at the intersection of the diagonals of the rectangle, into which the blade is inscribed (the blade contour line corresponding to its surface from the stator view, where 1, 2, 3, 4 are the contact points of the contour line and rectangle sides).

Two methods for radial clearances measuring between the blade tips of specified shape and turbine stator under the impeller axial displacements are proposed in [33]. Both methods are implemented by the “distributed cluster” of two SCECS. The SCECS are installed on a turbine shell with the angular shift of the centers of their SE at the distance of 1.5Δψ*_b_*, where Δψ*_b_* is the angular pitch of the blades (Table 2, Row 1). The methods differ in sensor placement relative to the conventional b.g.c., as well as the orientation of their SE in relation to the blade tips. Therefore, the methods have different sensitivities to the radial clearances and permissible ranges of shaft axial displacements.

The placement of the “distributed cluster” of SCECS on the turbine stator in accordance with the first of the considered methods is shown in Figure 8a. The sensors are represented by their SE which are shifted relative to the conventional b.g.c. in such a way that the electromagnetic interaction occurs between the SE and the tip near the blade outlet edge where the curvature and the thickness are significantly less than in the middle part and, especially, near the blade inlet edge [70]. The distance between the protrusions decreases with a decrease in blade thickness and, consequently, the “protrusions effect” on the inductance changes of SE_1_ and SE_2_ is reduced. Addition reduction of the “protrusions effect” is provided by SE_1_ and SE_2_ turning by an angle of 30 ÷ 60 degrees counterclockwise relative to the direction of the impeller rotation (over *z* axis). As a result, the signal “doubling” at the MC output due to the specificity of the blade tips shape was avoided (the signal is shown in Figure 8b). Further extraction of the signal’s informative component is carried out in the traditional way for SCECS, which was discussed above.

The desire to reduce the effects of blade airfoil curvature and the tips protrusions presence by the displacement of the SCECS’ SE to the outlet edge of the blade in conjunction with the SE turn (the SE are oriented almost “quasi-perpendicularly” to the contour line of the blade airfoil), leads to a significant decrease in useful changes of the SE equivalent inductances. As a result, the sensitivity to the monitored parameter (the radial clearance) is significantly low. This limits the method’s applicability, especially for high-powered engines where the changes in both clearances and shaft axial displacements can be large.

The second method [35] provides a higher sensitivity to the monitored parameters, but its implementation requires more complex processing of the MC output signals from “distributed cluster” of SCECS. In this case, SCECS are placed near the conventional b.g.c. and their SE are turned to local «quasi-parallelism» to the contour line of the blade airfoil (Figure 9a). At the same time, the effect of signal “doubling” at the MC output (Figure 9b) manifests itself to the maximum extent when the edges of the tip’s protrusions pass through the sensitivity zones of both SCESCS. However, this effect is not eliminated in Method 2, but on the contrary, it is assumed to be useful for evaluating the shaft axial displacements.

For this purpose, an approximate equality of *min*_1_*L*_SE1_, *min*_2_*L*_SE1_ and *min*_1_*L*_SE2_, *min*_2_*L*_SE2_ inductances when the blade tip protrusions pass through the sensitivity zones of both SCECS (Figure 9b, diagrams 1, 2) is achieved by adjusting the turn angle of sensors’ SE. The appearance of the shaft “negative” axial displacement involves the increase in the first of two minimum values of both SE_1_ and SE_2_ inductances (*min*_1_*L*_SE1_ < *min*_2_*L*_SE1_, *min*_1_*L*_SE2_ < *min*_2_*L*_SE2_) (Figure 9b, diagrams 3, 4). Moreover, with further shaft displacement in the same direction the signal “doubling” effect gradually disappears, and the nature of both SE inductance changes becomes identical to the diagrams in Figure 8b. Similarly, the shaft axial displacement in the “positive” direction leads to the increase in the second of the two minimum values of both SE_1_ and SE_2_ inductances (Figure 9b, diagrams 5, 6) with the subsequent disappearance of the signal “doubling” effect. In other words, there is a “competition” between minimum inductance values with a “leader” change at the point *x* = 0 (initial position). Therefore, Method 2 considers the smallest of the two inductance minimum values as an informative parameter. If both minimums are equal, then it does not matter which one is chosen for further coordinates (radial clearances and shaft axial displacements) calculation.

### 7.2. Protection from the Influence of the Adjacent Elements of the Monitored Object Design on the Result of SCECS Signals Conversion

The effect associated with the influence of adjacent structural elements of the monitored object on the SCECS signals conversion has been known for a long time. In [71] it is proposed to exclude the influence of the surrounding ECP metal on the measurement result, when the sensor is installed on the monitored object, by metal removing near the sensor’s end part. With this aim, the body of the monitored object is drilled into the sensor’s location zone in such a way that a minimum gap between the ECP body and the body of the monitored object of at least 2.5 mm is ensured. Systems on the base of SCECS initially used a slightly different approach to sensors installed on the monitored object excluding the influence of the surrounding SCECS metal. For these purposes, a special fastening element for the SCECS, the holder, is used. The holder, in fact, is an element of the monitored object design and its exact copy is used on a specialized installation (an example of such a device is presented in Figure 5) for calibration of the measuring channels with SCECS [28]. This ensures the identity of SCECS operating conditions during the sensor’s metrological certification and when it is installed at the monitored object.

Unfortunately, the practical applications of the measurement systems with SCECS under test bench conditions demonstrated that surrounding the SCECS metal at the place of the sensor’s installation on the body of the monitored object is not the only influencing factor of this group. In particular, the impact of adjacent blades on the results of the blade tip displacement measuring by SCECS was revealed in the tests of highly loaded stages of the GTE compressor and turbine [33,55]. Further, this effect was investigated on the basis of the models of electromagnetic interaction of the SCECS’s SE with the monitored and adjacent blades. The results of the study for the “distributed cluster” of SCECS are presented in [33,62]. For example, it is shown that reducing the pitch of the blades from 32 mm to 26 mm leads to increasing the effect of the adjacent blades by several times (up to 0.03% for the working sensor and 1.5% for the compensation sensor, i.e., the witness SCECS becomes sensitive to the movement of the monitored object and ceases to perform its compensatory functions properly). The smaller the radial gap to the monitored blade, the greater the adjacent blade’s effect.

A significant reduction of the possible errors due to the influence of adjacent blades can be achieved by replacing the existing calibration technique [33]. Unlike the well-known calibration method, the new one provides for the use of not a single, but several blades, which are installed in the calibration device (like the one shown in Figure 6) on a common basis with the same pitch in a linear term as on a real object. Essentially, such a design imitates a fragment of the impeller and provides the obtaining of the families of calibration characteristics (6) of measuring channels with SCECS considering the influence of the adjacent blades.

## 8. SCECS Sampling Synchronization Errors: Causes, Influence, and Ways to Reduce It

As noted earlier, the conversion of SCECS informative parameters (inductances) in many practical applications related to the position monitoring of the objects with a discrete surface (e.g., blade tips of compressor or turbine impellers, gear teeth or protrusions on measuring disks, etc.), should be performed at specified time points corresponding to strictly defined positions of the monitored object in the measurement zone. Traditionally, this problem is solved by preliminary measuring the speed of the object movement, further calculating the SCECS sampling moments, synchronizing the sampling beginning with a given object’s initial position, and, in fact, the formatting of the sensor’s pulse power supply at the calculated time points. The synchronization is usually carried out by industrial RPM sensors (e.g., DCHV-2500 [30,32], IS-445 [19]). Of course, the constant speed is assumed for all calculations.

At the same time, the production tolerances in the manufacture of individual elements of machines and mechanisms lead to the fact that, for example, blades or gear teeth may be at an unequal distance from each other and this causes the corresponding components of the measuring error. For instance, according to [72] the blade roots, which fix the blade into the turbine rotor, are made with an accuracy of ±0.01…±0.02 mm, and according to [73] the accuracy of the blade airfoil manufacturing is within ±0.03 mm. According to the estimates given in [74], the time error in SCECS power supply pulses generation during blade tips radial and axial displacement measuring can reach 6% under indicated deviations. It is proposed in [75] to correct the indicated error component by considering the individual design features of the monitored object. For these purposes, the real location of the machine elements is measured before the experiments start.

It should also be noted that the leading edge of the signal from the inductive PRM sensor changes with a change in the object speed. This results in a time shift of the entire sequence of SCECS power supply pulses. Ultimately, the error in determining the coordinates of monitored object displacements appears, although the accuracy of the period measuring remains at a high level. According to the estimates given in [74], the increase in leading edge pulse time by only 60 ns leads to the error in the SCECS channels up to 0.9%. The decrease in the edge duration of RPM signals conditioner output voltage or the application of SCECS with a modified signal converter as an RPM sensor [76] can reduce the specified error.

In addition, finally, in transient modes with fast speed changing of the monitored object moving through the SCECS measuring zone, the actual moments of the object appearing in the required position will be ahead or behind the calculated ones in which sensors’ informative parameters are fixed. As a result, the corresponding component of the measuring error appears. The error increases as the fixation moments shift from the time of synchro pulse occurrence. It is indicated in [74] regarding the task of radial and axial displacements monitoring of the compressor impeller blade tips, that the dynamic component of the error in SCECS power supply pulses generation varies from ±0.001% for the first blade up to ±17% for the 114th (last) blade under impeller speed drop from 3000 rpm to its shutdown with angular accelerations of 72 deg/s^2^. This effect can be significantly reduced by measuring the instantaneous velocities and accelerations of the monitored objects. In this case, there is no need to use additional RPM sensors. The required information can be obtained by special signals processing of the same SCECS that are used for the direct measurements of the coordinate displacements of the machines’ structural elements. For these purposes, the time of the extreme voltage values appear at the MC output when the adjacent elements pass through the sensitivity zone of the same SCECS is additionally fixed. The object’s speed is determined by the duration measuring of the time interval between neighboring extremes. The duration analysis of the adjacent time intervals allows for determining the object’s acceleration [77].

## 9. Conclusions

The measuring tasks that are solved by SCECS are often unique. The operation of this kind of sensor goes on in difficult and even extreme conditions under the influence of a large number of external factors that have a negative impact on the conversion result of the monitored parameters. The application of SCECS-based systems in industrial and bench conditions made it possible to identify most of these factors and determine the ways to, if not completely eliminate, then significantly reduce their impact. Of course, the review does not exhaust all possible influencing factors and the ways of their elimination, but it gives an overview of the main ones that can be called decisive. The effectiveness of the considered approaches and methods has been proven in laboratories and in bench tests of power plants for various purposes. At the same time, many of the proposed solutions are not limited only to SCECS-based systems. They can be successfully applied in measurement systems that use other types of ECP, as well as sensors built on other physical principles. A good example is publication [78], where the method of radial clearances measuring in GTE compressor on the base of radar sensors is considered and the influence of the shaft’s axial displacements is corrected in a manner similar to the considered in the review.

## 10. Patents

Patent SU, No. 1394912, 1986: High-temperature conductor eddy current converter.

Patent RF, No. 2272990, 2006: Method for measuring the multidimensional displacements and detecting the vibrations of the blade tips of a turbomachine rotor.

Patent RF, No. 2457432, 2012: Method for measuring radial clearances and axial displacements of the blades’ tips of turbine impeller.

Patent RF, No. 2525614, 2014: Device for measuring multi-coordinate displacements of the blades’ tips.

## Figures and Tables

**Figure 1 sensors-23-00351-f001:**
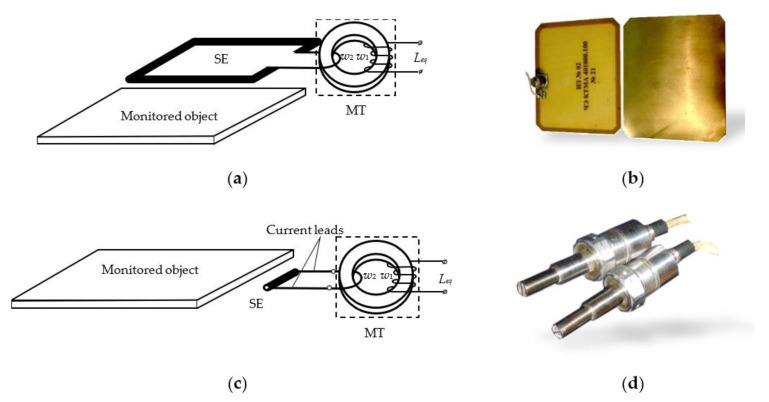
Schematic images (**a**,**c**) and the examples (**b**,**d**) of SCECS: the sensor with SE as a single current-carrying coil for measuring the gaps between the photocells of the adjacent flaps of the solar arrays (**a**,**b**) and the sensor with SE as a segment of a linear conductor for non-contact measuring of radial clearances in GTE compressor (**c**,**d**).

**Figure 2 sensors-23-00351-f002:**
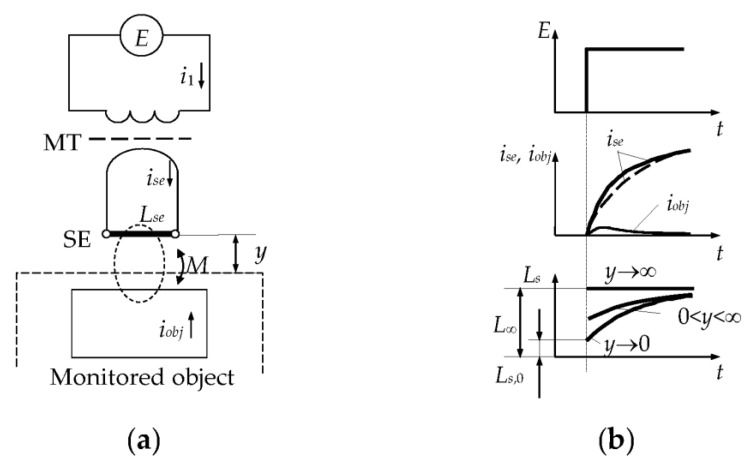
Double contour model of the electromagnetic interaction between the SE of the SCECS and the electrically conductive object (**a**), and the time diagrams of changes in the currents and equivalent inductance of the SE (**b**) [18,36].

**Figure 3 sensors-23-00351-f003:**
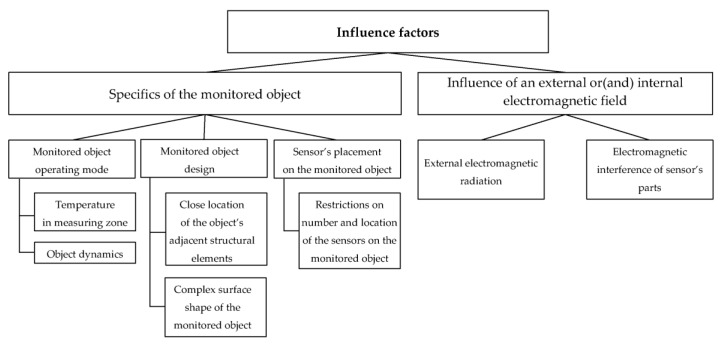
Classification of the factors affecting SCECS.

**Figure 4 sensors-23-00351-f004:**
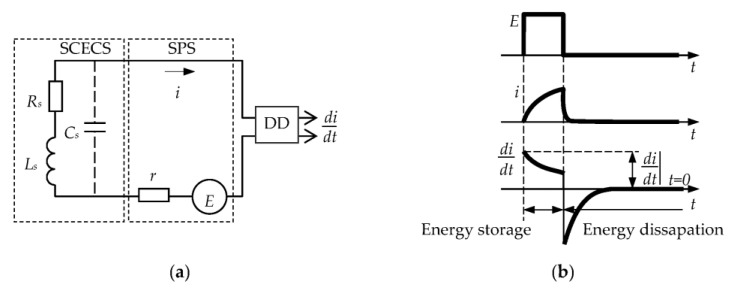
Equivalent scheme (**a**) and time diagrams (**b**) explaining the first derivative method.

**Figure 5 sensors-23-00351-f005:**
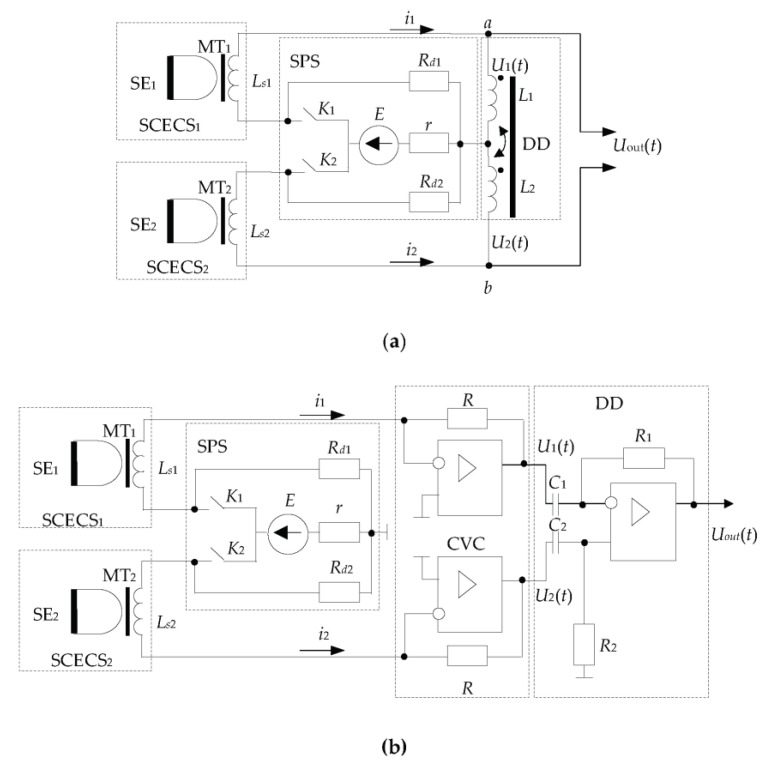
Measuring circuits with the differentiating devices on the base of Blumlein bridge (**a**) and operational amplifiers (**b**).

**Figure 6 sensors-23-00351-f006:**
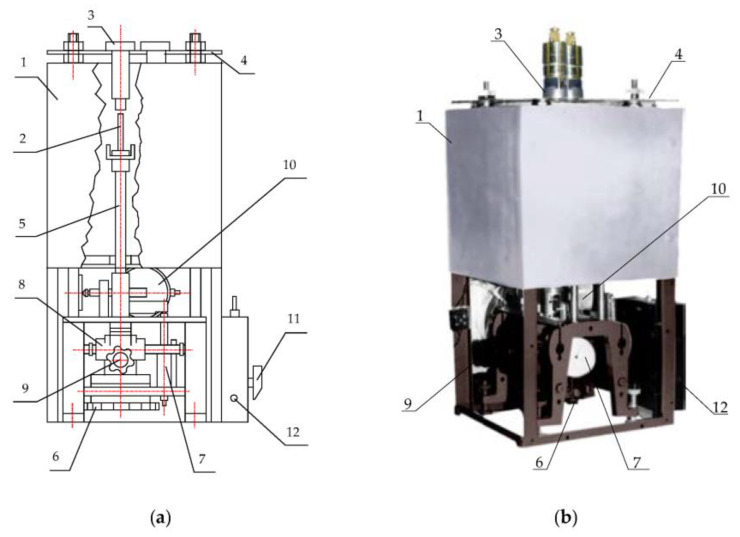
Special device for calibration of the measuring channels with SCECS: schematic view (**a**) and design of the device with working and compensative SCECS (**b**). 1—thermal chamber, 2—fragment of the monitored object, 3—bushing, 4—platform, 5—ceramic rod, 6—micrometric screw, 7—dial indicator, 8—carriage, 9—handwheel, 10—dial indicator, 11—regulator switch, 12—LED indicator.

**Figure 7 sensors-23-00351-f007:**
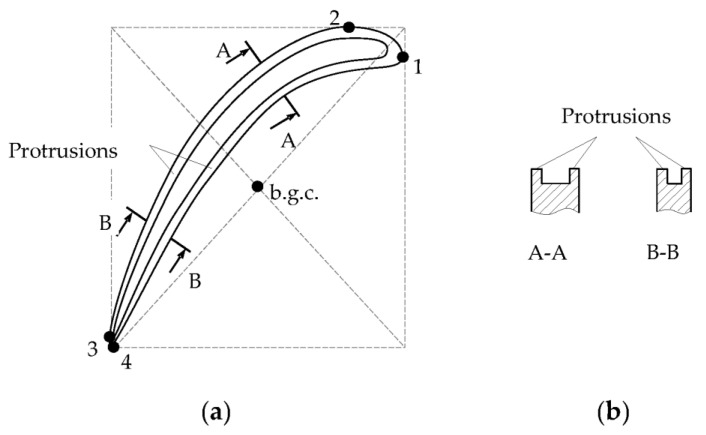
View of the turbine blade tip from the stator side: conditional blade’s geometric center (b.g.c.) (**a**) and its cross-sections A-A and B-B (**b**).

**Figure 8 sensors-23-00351-f008:**
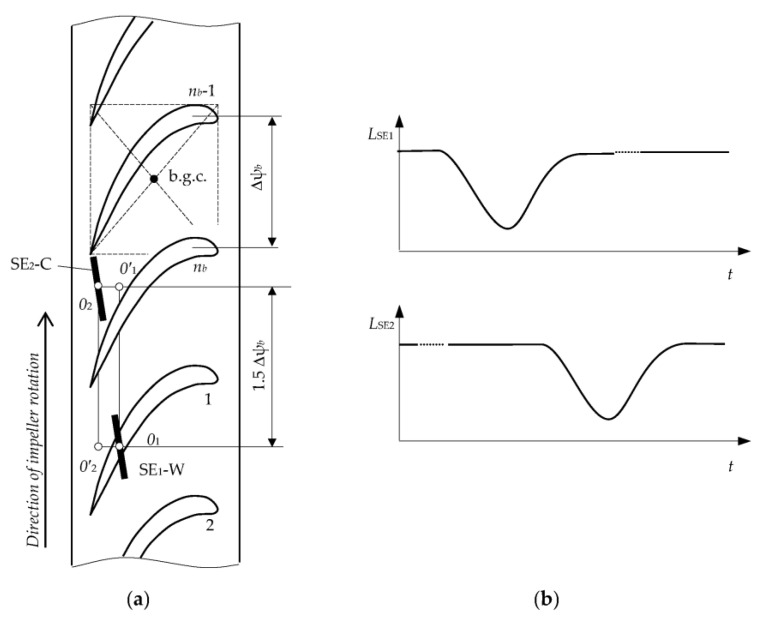
Method 1: placement of the “distributed cluster” of SCECS on a turbine stator (**a**) and sensors’ SE inductance changes in time (**b**).

**Figure 9 sensors-23-00351-f009:**
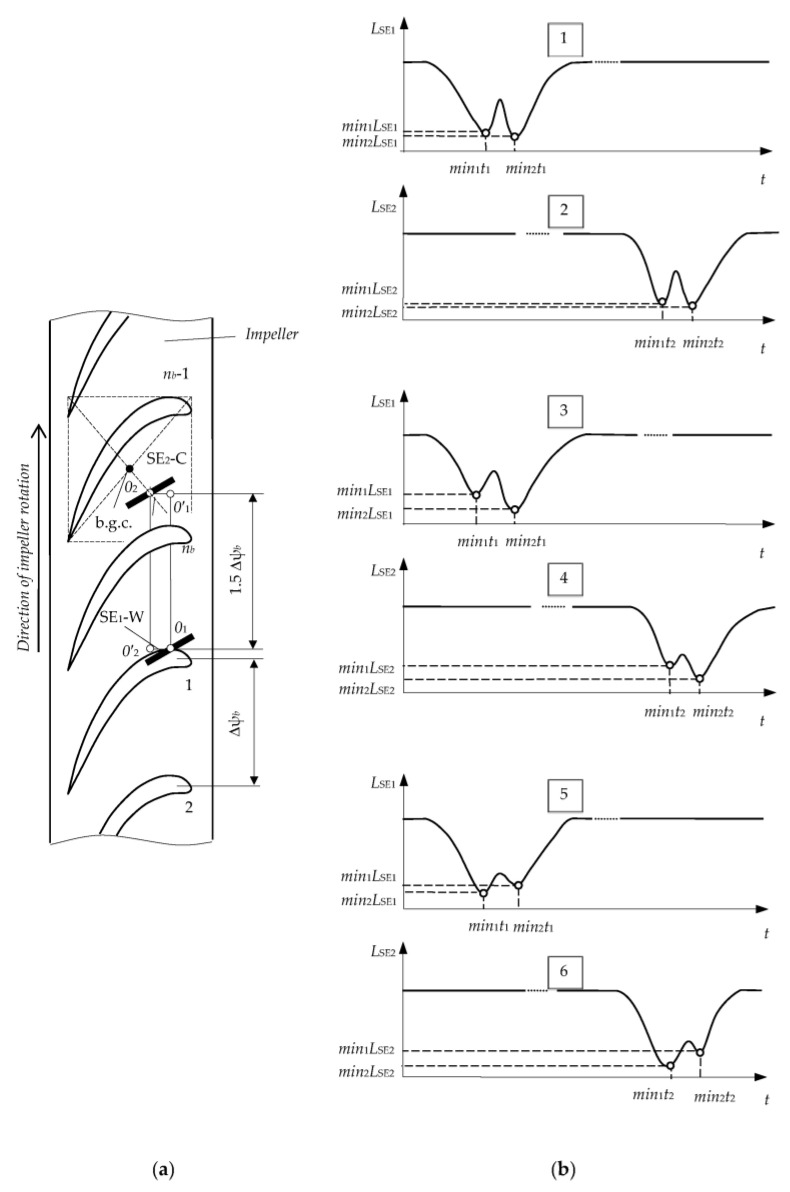
Method 2: placement of the “distributed cluster” of SCECS on a turbine stator (**a**) and sensors’ SE inductance changes in time (**b**).

**Table 1 sensors-23-00351-t001:** Methods for the discrete surfaces (blade tips) monitoring on the base of concentrated clusters of SCECS.

Sensorsper Cluster	Sensors (SE)Layout	Possible Measured Coordinates	Calibration Characteristic
1	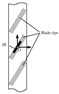	Axial displacement (*x*)	C1=f1(x,θ)
Radial clearance(*y*)	C1=f1(y,θ)
2	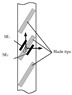	Axial and radialdisplacements(*x*, *y*)	{C1=f1(x,y,θ),C2=f2(x,y,θ).
3	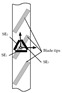	Axial, radial and twistdisplacements(*x*, *y*, *z*)	{C1=f1(x,y,z,θ),C2=f2(x,y,z,θ),C3=f3(x,y,z,θ).

**Table 2 sensors-23-00351-t002:** Methods for the discrete surfaces (blade tips) monitoring on the base of distributed clusters of SCECS.

Sensors per Cluster	Sensors (SE) Placement and Conversion Steps		
	Phase 1	Phase 2	Phase 3
2	** 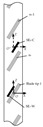 **	** 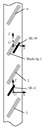 **	**-**
3	** 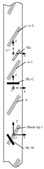 **	** 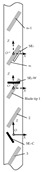 **	** 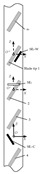 **

## Data Availability

The study did not report any data.

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
