# Peer review of "Reducing the Impact of Influence Factors on the Measurement Results from Single-Coil Eddy Current Sensors"

_sensors, 2022, doi:10.3390/s23010351_

Round 1

Reviewer 1 Report

Dear Authors,

The article submitted for review "Reducing the impact of influence factors on the measurement results from single-coil eddy current sensors" is a comprehensive overview of problems with using SCECS along with methods of dealing with influencing factors. The article consists of an introduction, a precise definition of the factors affecting SCECS, and subsequent chapters describing ways to minimize or compensate the affecting factors.
The article contains a lot of valuable information for both scientists and engineers. The article presents the topic in an understandable way, the language used is scientific, and the illustrative material used is a perfect complement to the article.
However, I think that before publishing the article, 2 major errors should be corrected:
- in chapter 2 there should also be a more detailed description of the operation of the sensors, so that the reader who is less familiar with the subject also understands the principle of operation of the SCECS sensors
- in the article there should be more references to up today works and authors should more often cite works by authors from America, Western Europe or Asia.
Minor bugs:
- authors giving trade names should write something more about the parameters of the material (e.g. line 163 "700NM ferrite cores")

So I think the article should be reconsider after major revision.

Author Response

Dear Reviewer,

the authors would like to express their sincere thanks for the constructive and precious comments that gave us a chance to improve the article. We carefully studied all the comments, questions, and suggestions and have revised the manuscript accordingly. Our responses in point-by-point manner are given below.

Point 1: in chapter 2 there should also be a more detailed description of the operation of the sensors, so that the reader who is less familiar with the subject also understands the principle of operation of the SCECS sensors

Response 1: Thank you for the comment. Working for a long time in any branch of knowledge, some things become self-evident. The authors fell into the same trap. You are certainly right, and it is important for the reader to know the SCECS’s operation principles, without which it is difficult to understand the proposed approaches and methods. We have expanded chapter 2 by adding Figure 2 and the detailed description of the operation of the sensor at the end of the section.

Point 2: in the article there should be more references to up today works and authors should more often cite works by authors from America, Western Europe or Asia.

Response 2: The authors thank you for these recommendations which will certainly improve our article. Of course, we are familiar with the works of our colleagues from Canada, USA, UK, Poland, Italy, India, and China at least. We have made additions to the text of the article with appropriate references ([14], [16,17], [22-25], [34]).

Point 3: authors giving trade names should write something more about the parameters of the material (e.g. line 163 "700NM ferrite cores")

Response 3: We have added some analogues of some materials that are probably more familiar to the article readers (e.g. line 202) and have deciphered (wrote the parameters) of ferrite core 700NM in line 371

We hope the manuscript is now suitable for publication.

Sincerely,
The authors.

Reviewer 2 Report

This review mainly summarizes the factors that affect the measurement results of single-coil eddy current sensors. The typical design of SCECS, the main influencing factors and the methods to reduce the degree of influence are introduced. Overall, the paper is interesting. In Reviewer's opinion, several issues should be addressed to improve this paper as follows:

1.      In the introduction, the description of eddy current probes is lengthy. Unnecessary descriptions can be removed.

2.      The last paragraph of the introduction focuses on the work of this review. However, it is completely duplicated with the abstract. The author can express it differently and describe the chapters of the article accordingly.

3.      The illustration in the figure does not meet the standards of sensors.

4.      In Figure 5, the specific part name can be marked.

5.      The writing of abstracts and conclusions does not highlight the focus of the review. The author can re-write according to the general writing method of abstracts and conclusions.

6.      The logic of the article is not clear, the author needs to revise.

7.      Please check the reference format.

Author Response

Dear Reviewer,

the authors would like to express their sincere thanks for the constructive and precious comments that gave us a chance to improve the article. We carefully studied all the comments, questions, and suggestions and have revised the manuscript accordingly. Our responses in point-by-point manner are given below.

Point 1: In the introduction, the description of eddy current probes is lengthy. Unnecessary descriptions can be removed.

Response 1: We have shortened the description of the eddy current probes in the introduction, excluding details that are insignificant in our opinion.

Point 2: The last paragraph of the introduction focuses on the work of this review. However, it is completely duplicated with the abstract. The author can express it differently and describe the chapters of the article accordingly.

Response 2: The authors thank you for these recommendations. The abstract and the last paragraph of the introduction were significantly adjusted in accordance with the suggestions of the respected reviewer.

Point 3: The illustration in the figure does not meet the standards of sensors.

Response 3: The respected reviewer unfortunately did not specify the figure number. The authors assumed that it was Figure 1 and made the appropriate corrections.

Point 4: In Figure 5, the specific part name can be marked.

Response 4: We have added the specific part names in Figure 5 (Figure 6 in new version) according to the recommendation.

Point 5: The writing of abstracts and conclusions does not highlight the focus of the review. The author can re-write according to the general writing method of abstracts and conclusions.

Point 6: The logic of the article is not clear, the author needs to revise.

Responses 5, 6: The abstract, the introduction and the conclusion sections were adjusted in accordance with the suggestions of the respected reviewer.

Point 7: Please check the reference format.

Response 7: We have checked the references according to the Sensors Template.

We hope the manuscript is now suitable for publication.

Sincerely,
The authors

Round 2

Reviewer 1 Report

The authors have introduced the corrections suggested by me, I believe that the article is ready for publication.